Microbiology
Spectrum
# Arabinosyltransferase C Mediates Multiple Drugs Intrinsic Resistance by Altering Cell Envelope Permeability in *Mycobacterium abscessus*

Shuai Wang,[a,b,c,d,e] Xiaoyin Cai,[b,c,d] Wei Yu,[b,c,d,e] Sheng Zeng,[b,d] Jingran Zhang,[b,d,e,f] Lingmin Guo,[b,c,d] Yamin Gao,[b,c,d,e] Zhili Lu,[b,c,d,e] H. M. Adnan Hameed,[b,c,d,e] Cuiting Fang,[b,c,d,e] Xirong Tian,[b,c,d,e] Buhari Yusuf,[b,c,d,e] Chiranjibi Chhotaray,[b,c,d] M. D. Shah Alam,[b,c,d,e] Buchang Zhang,[g] Honghua Ge,[g] Dmitry A. Maslov,[h] Gregory M. Cook,[i,j] Jiacong Peng,[d,k] Yongping Lin,[d,k] Nanshan Zhong,[d,k,l] Guoliang Zhang,[a] Tianyu Zhang[b,c,d,e]

aNational Clinical Research Center for Infectious Diseases, Guangdong Provincial Clinical Research Center for Tuberculosis, Shenzhen Third People's Hospital, Shenzhen, China

bState Key Laboratory of Respiratory Disease, Guangzhou Institutes of Biomedicine and Health, Chinese Academy of Sciences, Guangzhou, China

cUniversity of Chinese Academy of Sciences, Beijing, China

dGuangdong-Hong Kong-Macao Joint Laboratory of Respiratory Infectious Diseases, Guangzhou Institutes of Biomedicine and Health, Chinese Academy of Sciences, Guangzhou, China

eChina-New Zealand Joint Laboratory on Biomedicine and Health, Guangzhou, China

fSchool of Life Sciences, University of Science and Technology of China, Hefei, Anhui, China

gInstitutes of Physical Science and Information Technology, Anhui University, Hefei, China

hLaboratory of Bacterial Genetics, Vavilov Institute of General Genetics, Russian Academy of Sciences, Moscow, Russia

iDepartment of Microbiology and Immunology, School of Biomedical Sciences, University of Otago, Dunedin, New Zealand

jMaurice Wilkins Centre for Molecular Biodiscovery, The University of Auckland, Private Bag, Auckland, New Zealand

kState Key Laboratory of Respiratory Disease, National Clinical Research Center for Respiratory Disease, The National Center for Respiratory Medicine, The First Affiliated Hospital of Guangzhou Medical University, Guangzhou, China

lGuangzhou Laboratory, Bio-Island, Guangzhou, China

**ABSTRACT** *Mycobacterium abscessus* is an emerging human pathogen leading to significant morbidity and even mortality, intrinsically resistant to almost all the antibiotics available and so can be a nightmare. Mechanisms of its intrinsic resistance remain not fully understood. Here, we selected and confirmed an *M. abscessus* transposon mutant that is hypersensitive to multiple drugs including rifampin, rifabutin, vancomycin, clofazimine, linezolid, imipenem, levofloxacin, cefoxitin, and clarithromycin. The gene *MAB_0189c* encoding a putative arabinosyltransferase C was found to be disrupted, using a newly developed highly-efficient strategy combining next-generation sequencing and multiple PCR. Furthermore, selectable marker-free deletion of *MAB_0189c* recapitulated the hypersensitive phenotype. Disruption of *MAB_0189c* resulted in an inability to synthesize lipoarabinomannan and markedly enhanced its cell envelope permeability. Complementing *MAB_0189c* or *M. tuberculosis embC* restored the resistance phenotype. Importantly, treatment of *M. abscessus* with ethambutol, a first-line antituberculosis drug targeting arabinosyltransferases of *M. tuberculosis*, largely sensitized *M. abscessus* to multiple antibiotics *in vitro*. We finally tested activities of six selected drugs using a murine model of sustained *M. abscessus* infection and found that linezolid, rifabutin, and imipenem were active against the *MAB_0189c* deletion strain. These results identified MAB_0189 as a crucial determinant of intrinsic resistance of *M. abscessus*, and optimizing inhibitors targeting MAB_0189 might be a strategy to disarm the intrinsic multiple antibiotic resistance of *M. abscessus*.

**IMPORTANCE** *Mycobacterium abscessus* is intrinsically resistant to most antibiotics, and treatment of its infections is highly challenging. The mechanisms of its intrinsic resistance remain not fully understood. Here we found a transposon mutant hypersensitive to a variety of drugs and identified the transposon inserted into the *MAB_0189c* (orthologous *embC* coding arabinosyltransferase, EmbC) gene by using a newly developed rapid

Address correspondence to Tianyu Zhang, zhang_tianyu@gibh.ac.cn, or Guoliang Zhang, szdsyy@aliyun.com.

The authors declare no conflict of interest.

and efficient approach. We further verified that the *MAB_0189c* gene played a significant role in its intrinsic resistance by decreasing the cell envelope permeability through affecting the production of lipoarabinomannan in its cell envelope. Lastly, we found the arabinosyltransferases inhibitor, ethambutol, increased activities of nine selected drugs *in vitro*. Knockout of *MAB_0189c* made *M. abscessus* become susceptible to 3 drugs in mice. These findings indicated that potential powerful *M. abscessus* EmbC inhibitor might be used to reverse the intrinsic resistance of *M. abscessus* to multiple drugs.

**KEYWORDS** *Mycobacterium abscessus*, intrinsic resistance, *MAB_0189c*, permeability

Mycobacterium abscessus is a rapidly-growing environmental *mycobacterium* causing local or disseminated lung, skin, and soft tissue infections in patients with underlying conditions such as cystic fibrosis and chronic obstructive pulmonary diseases (1). It is considered as an "incurable nightmare" due to its high-level intrinsic resistance to most antibiotics commonly used for bacterial infections, in particular, including nearly all antituberculosis drugs (2). Current therapeutic options, as recommended by the Infectious Disease Society of America, comprise the combination of amikacin (AMK), cefoxitin (CEF), imipenem (IMP), and macrolides (3). However, partly due to the lack of appropriate treatment options, the average treatment success rate is only ∼ 58% even with very long treatment duration (4). Thus, efforts to identify new therapeutic regimens should be emphasized.

Intrinsic resistance of *M. abscessus* to antibiotics is mediated by multiple factors. First, *M. abscessus* is known to produce numerous enzymes that can modify either the drug targets or the drugs themselves, leading to reduced antimicrobial effects. This is exemplified by the expression of ribosomalmethylase encoded by *erm*(41) leading to macrolide resistance, rifampin (RIF)-inactivating ADP-ribosyltransferase, and $\beta$-lactam-hydrolyzing $\beta$-lactamase Bla$_{Mab}$ (5–7). Second, genetic polymorphisms also play key roles in determining intrinsic resistance of *M. abscessus*. For instance, polymorphism in *atpE* confers resistance of *M. abscessus* to bedaquiline, a recently approved drug with high potency against *M. tuberculosis* (8). In addition to these mechanisms, the relatively low permeability of the cell envelope coupled with the presence of multidrug export systems contributes to the intrinsic antibiotic resistance by limiting intracellular concentrations of drugs (2, 9, 10). A recent study even demonstrated that enhanced cell envelope permeability resulting from compromised glycosylation of a lipoprotein renders *M. abscessus* more sensitive to $\beta$-lactams, vancomycin (VAN), and RIF (10).

A well-versed understanding of the genetic basis of intrinsic antibiotic resistance is prerequisite for the discovery and development of synergistic drug combinations, where one agent can restore the activity of a failing antibiotic by disrupting one of the above-mentioned mechanisms. A relatively good example in this regard is the identification of avibactam, a potent inhibitor of *M. abscessus* $\beta$-lactamase, to efficiently facilitate the activities of several $\beta$-lactams by 4- to 32-fold when used in combination against *M. abscessus* (11, 12).

Forward genetics approaches such as transposons (Tn) mutagenesis have been widely applied to generate random insertion mutants for subsequent identification of those with altered phenotypes. Antibiotic susceptibility screening of large collections of Tn mutants in combination with a Tn-Seq approach has the potential to identify mechanisms and factors associated with antibiotic resistance, which may further facilitate the development of new drugs. It has been recently applied in the discovery of a repertoire of previously unknown factors associated with resistance to oxazolidinones, fluoroquinolones, and aminoglycosides in pathogens such as *Staphylococcus aureus*, *Pseudomonas aeruginosa* and *Escherichia coli* (13–15).

Here, we generated a Tn mutant library of *M. abscessus* for screening hypersensitive mutants and identified rapidly and efficiently Tn insertion sites in target clones using a newly developed highly efficient method combining next generation sequencing (NGS) and multiple PCR. This allowed us to identify the *MAB_0189c*, a gene encoding a probable arabinosyltransferase C (EmbC) responsible for the polymerization of arabinose into the arabinan of lipoarabinomannan (LAM), as an important player in determining resistance of

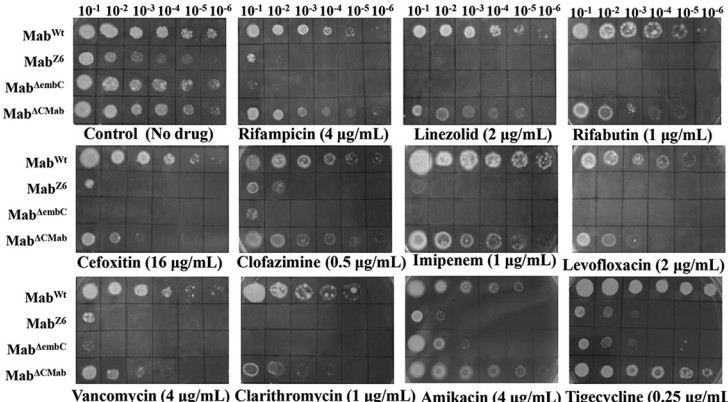

**FIG 1** Differences in antibiotic sensitivities of *M. abscessus* strains. Tenfold serial dilutions of *M. abscessus* strains grown to $OD_{600}$ of 0.7 were spotted on Middlebrook 7H10 containing indicated concentrations of antibiotics. Plates were incubated for 3 days. Representative data from 3 independent experiments are shown.

*M. abscessus* to multiple antimicrobials. Disruption of the *MAB_0189c* gene markedly enhanced cell envelope permeability, which explained well the enhanced antibiotic susceptibilities to multiple drugs both *in vitro* and *in vivo*. Finally, we showed that ethambutol (EMB), a first-line antituberculosis drug inhibiting the arabinosyltransferases of *M. tuberculosis*, remarkably sensitized *M. abscessus* to a panel of existing antimicrobials.

## RESULTS

**Identification of a Tn mutant hypersensitive to multiple drugs.** Using *Himar1* mutagenesis (16), we first constructed a Tn mutant library containing approximately $1.25 \times 10^5$ mutant clones of *M. abscessus* GZ002 (Mab^Wt), a clinical strain isolated in China and verified by 16s RNA gene sequencing and whole-genome sequencing (17). To identify genes associated with intrinsic antibiotic resistance of *M. abscessus*, we determined the growth of >5,000 randomly selected mutants on agar plates containing subinhibitory concentrations of RIF (4 $\mu$g/mL). This initial screening led to the identification of 10 mutants hypersensitive to RIF with 1 designated as Mab^Z6 (Fig. 1). In 7H9 liquid medium, the MIC of RIF against the Mab^Z6 strain was 1/32 of that of Mab^Wt (Table 1), confirming the hypersensitive phenotype. The susceptibilities of Mab^Wt and Mab^Z6 to other commonly used antibiotics were further tested. Interestingly, the Mab^Z6 strain was also more susceptible to linezolid (LIN), rifabutin (RFB), CEF, clofazimine (CLF), IMP, levofloxacin (LEV), VAN, and clarithromycin (CLA). However, on tigecycline (TGC) and AMK it was found partially less sensitive compared to other drugs (Fig. 1). In addition, the MIC of Mab^Z6 to TGC and AMK in liquid culture was observed similar to that of Mab^Wt, whereas Mab^Z6 was hypersensitive to other drugs tested (Table 1). The MICs to Mab^Z6 decreased to 1/2 ~1/64 compared to those to Mab^Wt. The detailed functional description of other transposon mutants is provided in Table S1. Most of the transposon disrupted genes belong to different functional categories, and the prime objective of the current study is to determine the key factor involved in intrinsic resistance in Mab^Z6.

**Disruption of *MAB_0189c* in Mab^Z6.** We developed a new method to efficiently identify the Tn insertion sites in the mutants (Fig. S1A in the supplemental material). The genomic DNAs mixture of 10 Tn mutants was subject to NGS, allowing for the mapping of 10 insertion sites in only one NGS reaction. Subsequently, multiple PCRs and sequencing were performed to identify the individual insertion site for each mutant. Using this efficient approach, we identified the Tn-disrupted gene in the Mab^Z6 strain to be the *EFV83_RS12000* gene of Mab^Wt (= the *MAB_0189c* of *M. abscessus* ATCC 19977) (Fig. S1B and S1C) (17). The Tn inserted into the "TA" dinucleotide located 1,413 bases downstream of the starting codon of the 3,258-bp-long *MAB_0189c* (Fig. S1B).

**TABLE 1** MICs of various drugs determined in 7H9 medium[a]

| Antibiotics[b] | *M. abscessus* strains/MICs ($\mu$g/mL) | | | | |
|---|---|---|---|---|---|
| | Mab$^{Wt}$ | Mab$^{Z6}$ | Mab$^{\Delta embC}$ | Mab$^{\Delta CMab}$ | Mab$^{\Delta CMtb}$ |
| Clarithromycin | >32 | 2 | 2 | >32 | >32 |
| Clofazimine | 2-4 | 1 | 0.5 | 2 | 2 |
| Tigecycline | 1 | 1 | 1 | 1 | 1 |
| Vancomycin | 128 | 2 | 2 | 128 | 128 |
| Amikacin | 8 | - | 8 | | |
| Rifampin | 128 | 4 | 4 | 128 | >128 |
| Imipenem | 16 | 1 | 1 | 16 | 16 |
| Linezolid | 64 | 2 | 2 | 64 | 32 |
| Levofloxacin | 64 | 2 | 2 | 32–64 | 32–64 |
| Cefoxitin | 64 | 16 | 16 | 32 | 16 |
| Rifabutin | 16 | 1 | 1 | 16 | 8 |

[a]Broth microdilution method was used to determine the MICs. The MIC was defined as the lowest drug concentration that prevented visible bacterial growth. The experiment was performed in triplicate and repeated twice.
[b]Drugs for *in vivo* study are underlined.

**Deletion of *MAB_0189c* resulted in hypersensitivity to multiple antibiotics *in vitro*.** The amino acid sequence of MAB_0189 showed 68.40% identity to *M. tuberculosis* EmbC and 68.94% identity to *M. smegmatis* EmbC, indicating the role of MAB_0189 can be EmbC in *M. abscessus* (Fig. S2). A selectable marker-free, isogenic deletion of *MAB_0189c* in Mab$^{Wt}$ was constructed using recombineering and the Xer/*dif* system as described previously (18, 19) and designated as Mab$^{\Delta embC}$ (Fig. S3). Mab$^{\Delta embC}$ was complemented with pMV261-MAB_0189c expressing *MAB_0189c* or pMV261-embC$_{Mtb}$ expressing *M. tuberculosis embC* under the strong mycobacterial promoter *hsp60*, to obtain strains Mab$^{\Delta CMab}$ and Mab$^{\Delta CMtb}$ respectively. Mab$^{\Delta embC}$ grew slower than its parent strain Mab$^{Wt}$ in Middlebrook 7H9 medium, a phenotype partially restored in the complemented strain (Fig. S3D). However, Mab$^{\Delta embC}$ attained a similar growth peak to that of Mab$^{Wt}$ after 72 h of growth (Fig. S3D).

To confirm the role of the *MAB_0189c* in intrinsic antibiotic resistance in *M. abscessus* further, we tested the antibiotic susceptibilities of Mab$^{Wt}$, Mab$^{\Delta embC}$, Mab$^{\Delta CMab}$, and Mab$^{\Delta CMtb}$ *in vitro* first. Corroborating the phenotype of the Tn insertion strain Mab$^{Z6}$, Mab$^{\Delta embC}$ exhibited a markedly enhanced sensitivity to RIF, CLR, CLF, VAN, IMP, LIN, LEV, CEF, and RFB compared to Mab$^{Wt}$ (Table 1 and Fig. 1). Expression of *MAB_0189c* in the complemented strain recovered the antibiotic resistance comparable to the resistance levels of Mab$^{Wt}$ (Table 1 and Fig. 1), thus confirming that *MAB_0189c* is critical for the intrinsic antibiotic resistance in *M. abscessus*. Importantly, expression of *M. tuberculosis embC* also restored *M. abscessus* drug resistance to the Mab$^{\Delta CMab}$ levels (Table 1). Hence, based on mycobrowser (https://mycobrowser.epfl.ch/), the identity comparison, and the complementation experiments above, we propose that the function of MAB_0189 is the same as EmbC, i.e., participating in the synthesis of LAM as an arabinosyltransferase C.

**Increased cell envelope permeability in Mab$^{\Delta embC}$.** In *M. tuberculosis*, EmbC is one of the three arabinosyltransferases required for the biosynthesis of LAM, an important structural component of mycobacterial cell envelope (20). Therefore, the effect of *MAB_0189c* on LAM was observed by sodium dodecyl sulfate (SDS)–polyacrylamide gel electrophoresis (PAGE) analysis. In agreement with a previous report (20), Mab$^{\Delta embC}$ strain retains lipomannan (LM) synthesis but is deficient in LAM (Fig. 2A, lane2), and complementation largely restored the normal phenotype (Fig. 2A, lane 3). Disruption of the LAM in *M. smegmatis* and *Mycobacterium neoaurum* was previously shown to increase cell envelope permeability (21, 22). Therefore, we reasoned that the hypersensitivity of Mab$^{\Delta embC}$ to multiple antibiotics was due to an increase in cell envelope permeability. To confirm this hypothesis, we observed that Mab$^{\Delta embC}$ accumulated ethidium bromide in a larger amount compared to Mab$^{Wt}$. Complementation of *MAB_0189c* partially but significantly reduced the dye accumulation (Fig. 2B). We further compared the sensitivity of these strains to SDS,

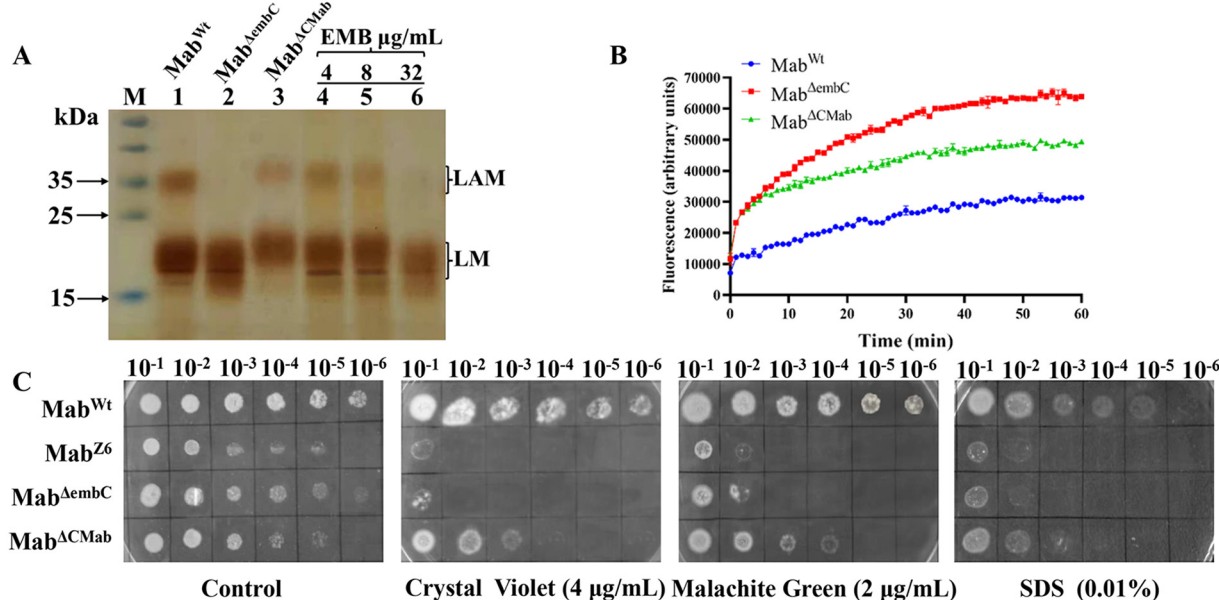

**FIG 2** Disruption of *MAB_0189c* in *M. abscessus* increases cell envelope permeability. (A) SDS/PAGE analysis of LAM from different strains. Lane M: protein MW standards. Lane 1: LAM and LM fraction from Mab$^{Wt}$. Lane 2: LAM and LM fraction from Mab$^{\Delta embC}$. Lane 3: LAM and LM fraction from Mab$^{\Delta CMab}$. Lanes 4–6: LAM and LM fraction from Mab$^{Wt}$ when culturing bacteria adding the concentration of EMB at 4 $\mu$g/mL, 8 $\mu$g/mL, and 32 $\mu$g/mL. (B) Accumulation of ethidium bromide in different strains. (C) Sensitivity of *M. abscessus* strains to malachite green, crystal violet, and SDS. The experiments were performed at least thrice, and only one representative image is shown in each case.

malachite green, and crystal violet. Assessment of sensitivity to these compounds was widely used as an alternative methodology to determine cell envelope permeability (23). Consistent with the ethidium bromide uptake assay, both Mab$^{\Delta embC}$ and Mab$^{Z6}$ were found to be more sensitive to the three compounds than Mab$^{Wt}$. The complemented strain partially restored resistance to the compounds (Fig. 2C). Together, these results point to an increased cell envelope permeability of Mab$^{\Delta embC}$.

**EMB sensitizes *M. abscessus* to multiple antibiotics.** Given the important role of *MAB_0189c* in intrinsic resistance of *M. abscessus* to multiple drugs (Table 1), we tested the possibility of sensitizing this bacterium by chemical inhibition of MAB_0189. The homologous protein of MAB_0189 (i.e., EmbC) is known to be targeted by EMB, a first-line antituberculosis drug (24). We therefore tested whether EMB could increase the antibiotic susceptibility of Mab$^{Wt}$ as well as 3 other randomly selected *M. abscessus* clinical isolates (Table 2). *M. abscessus* has been reported naturally resistant to EMB, which was supposed be due to amino acid difference in EmbB (another target of EMB) between *M. abscessus* and *M. tuberculosis* (25). The MICs of EMB against the selected *M. abscessus* strains exceeded to 64 $\mu$g/mL due to substitution of glutamine at position 281instead of isoleucine (I281Q) in EmbB in all isolates in this study (Fig. S4), as has been reported previously (25). Despite this, addition of sub-MICs of EMB (i.e., 4 and 8 $\mu$g/mL) resulted in a 2- to 8- fold decrease in MICs of RIF, CLR, CLF, VAN, IMP, LIN, LEV, CEF, and RFB (Table 2). In contrast, *M. abscessus* susceptibility to TGC and AMK did not change in the presence of EMB. We also detected the effect of EMB inhibition on LAM biosynthesis in *M. abscessus*. There were no significant changes in the content of LAM at 4 and 8 $\mu$g/mL of EMB (Fig. 2A, lanes 4–5), which may be because these concentrations of EMB are not enough to inhibit the synthesis of LAM efficiently as EMB cannot inhibit the growth of *M. abscessus* at such low concentrations. However, when we increased the concentration of EMB to 32 $\mu$g/mL, which is still much lower than its MIC to *M. abscessus*, we observed the clear inhibition of LAM synthesis (Fig. 2A, lane 6), indicating that EmbC activity was inhibited directly.

**Deletion of *MAB_0189c* sensitizes *M. abscessus* to multiple antibiotics *in vivo*.** CLR, CLF, VAN, IMP, LIN, and RFB showed much lower MICs to Mab$^{\Delta embC}$ than to Mab$^{Wt}$,

**TABLE 2** Susceptibility of Mab GZ002 and 3 isolates of *M. abscessus* to combinations of two concentrations of ethambutol (EMB)[a]

| Antibiotics | *M. abscessus* strains/concentrations (µg/mL) of EMB and other antibiotics | | | | | | | | | | | |
|---|---|---|---|---|---|---|---|---|---|---|---|---|
| | Mab^Wt^ (S) | | | Mab M1 (R) | | | Mab M3 (S) | | | Mab M4 (R) | | |
| | 0 | 4 | 8 | 0 | 4 | 8 | 0 | 4 | 8 | 0 | 4 | 8 |
| Clarithromycin | >32 | >32 | >32 | 1 | 0.5 | 0.25 | 0.5 | 0.25 | 0.125 | 1 | 0.5 | 0.25 |
| Clofazimine | 2–4 | 2 | 2 | 2 | 2 | 1 | 4 | 2–4 | 2 | 2 | 2 | 2 |
| Tigecycline | 1 | 1 | 1 | 1 | 1 | 1 | 8 | 8 | 4 | 2 | 2 | 2 |
| Vancomycin | 128 | 32 | 16 | >128 | >128 | >128 | >128 | >128 | >128 | >128 | >128 | >128 |
| Amikacin | 8 | 8 | 8 | >128 | >128 | >128 | 16 | 16 | 16 | 64 | 64 | 64 |
| Rifampin | 128 | 64 | 32 | 64 | 32 | 8 | >128 | >128 | >128 | 64 | 32 | 16 |
| Imipenem | 16 | 8 | 4 | 32 | 32 | 16 | 128 | 64 | 64 | 32 | 16 | 16 |
| Linezolid | 64 | 16 | 8 | 64 | 32 | 8–16 | 128 | 64 | 16 | 32 | 32 | 8 |
| Levofloxacin | 64 | 32 | 16 | 8 | 8 | 4 | 128 | 128 | 64 | 8 | 8 | 4 |
| Cefoxitin | 64 | 32 | 16 | 32 | 16 | 16 | 32 | 32 | 16 | 32 | 32 | 16 |
| Rifabutin | 16 | 8 | 4 | 4 | 2 | 1 | 32 | 16 | 16 | 4 | 2 | 1 |

[a]S, smooth; R, rough; Mab, *M. abscessus*. The experiment was performed in triplicate and repeated twice.

and the MICs of these drugs were also lower than blood peak concentrations in mice at normal dosage (26–29). We investigated their activities in mice infected with Mab^Wt^, Mab^ΔembC^, and Mab^ΔCMab^ strains. Consistent with the data obtained from the *in vitro* study, the drugs did not significantly decrease bacterial burden in lungs of mice infected with Mab^Wt^ or Mab^ΔCMab^ (Fig. 3B). However, in Mab^ΔembC^ infected mice, IMP, LIN, and RFB significantly decreased bacterial burden in lungs compared with sodium carboxymethyl cellulose (CMC-Na), the solvent control group ($P < 0.05$), at clinical relevant doses, while the other three drugs selected did not show obvious activities against Mab^ΔembC^ ($P > 0.05$) (Fig. 3B). Hence, IMP, LIN, and RFB were active against Mab^ΔembC^ *in vivo* at the tested doses.

Interestingly, bacterial burden in lungs of mice treated with CMC-Na in the Mab^ΔembC^ group at the treatment completion was significantly lower than that at treatment initiation ($P < 0.001$), while in the case of the Mab^Wt^ group, the lung CFU at treatment initiation and completion were very close ($P > 0.05$) (Fig. 3B). The Mab^ΔembC^ load of mice decreased even in the CMC-Na group, which indicated its virulence attenuated compared to Mab^Wt^. Unexpectedly, the bacterial burden in mice infected with the complemented strain was also decreased in the CMC-Na group ($P < 0.01$), thus suggesting that expression of *MAB_0189c* could not fully restore its function for the virulence *in vivo*.

We further assessed the *in vivo* efficacy of the drug combinations containing EMB plus IMP, CLF, RFB, or CLA against Mab^Wt^ in the same mouse model. Combining EMB with any drug selected did not enhance the drug activity *in vivo* compared to each drug alone (Fig. 3C). EMB may not penetrate inside *M. abscessus* enough to reach its target.

## DISCUSSION

*M. abscessus* is emerging as an important opportunistic pathogen responsible for severe infections (30, 31), having limited therapeutics options (32, 33). There is an urgent need for a better understanding of mechanisms of its intrinsic resistance to drugs available for new use of old drugs and discovery of potential new drugs. In this study, *MAB_0189c* was found to play a key role in intrinsic resistance to many drugs by maintaining cell envelope impermeability in *M. abscessus*.

Here we proved that *MAB_0189c*-encoded EmbC as the *embC* gene from *M. tuberculosis* could complement Mab^ΔembC^ for the drug susceptibilities (34). Interestingly, deletion of *embC* is tolerated in the fast-growing *M. smegmatis* (35), while its disruption is lethal in *M. tuberculosis* (36, 37). These observations indicate the differences in tolerability toward LAM defects between the two mycobacterial species. In this study, we demonstrated for the first time that *embC* (*MAB_0189c*) is nonessential for viability in *M. abscessus*, which could be another main reason why *M. abscessus* is intrinsic to EMB even though EmbC is the target of EMB (24).

In this study, we found EMB increased the sensitivities of *M. abscessus* to RIF, VAN, CLF, LIN, RFB, IMP, LEV, CEF and CLA, which was very similar to that observed in the

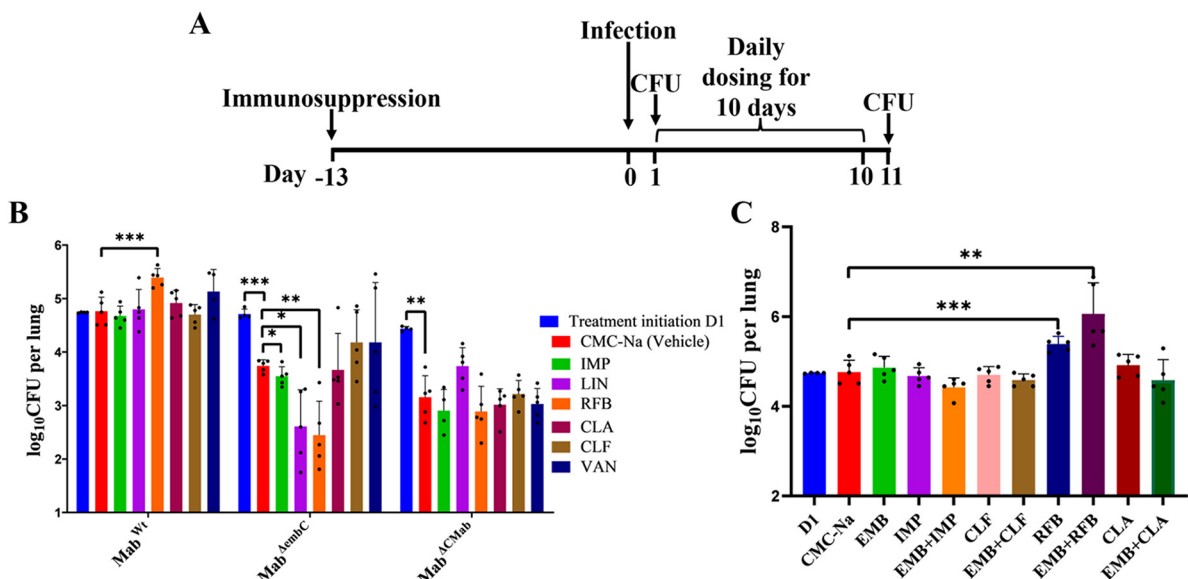

**FIG 3** Assessment of susceptibility to antibiotics *in vivo*. (A) Schematic representation of the murine *M. abscessus* lung infection model used in this study. All mice were immunosuppressed with DEXA prior to infection. At designated time points, lungs of mice were homogenized and plated on agar for CFU determination. (B) *M. abscessus* CFU in the lungs of mice treated with antibiotic versus controls. (C) *M. abscessus* CFU in the lungs of mice treated with antibiotic combinations versus single antibiotics versus untreated controls. Drugs were administered at the following doses (mg/kg): CLR 250, CLF 50, LIN 100, RFB 20, VAN 110, IMP 100, EMB 200, and 0.4% (wt/vol) CMC-Na control. Data points represent individual mice. *, $P < 0.05$; **, $P < 0.01$; ***, $P < 0.001$.

Mab$^{\Delta embC}$ strain. Moreover, EMB also showed the interference in the synthesis of LAM here. This may be due to the inhibition of EmbC but not by the disturbance in the production of arabinogalactan through inhibiting EmbA/EmbB of *M. abscessus*. To our knowledge, this is the first study identifying the EmbC as a potentially viable drug target for counteracting the intrinsic antibiotic resistance of *M. abscessus*. Novel potent inhibitors targeting MAB_0189 (EmbC) might make multiple inactive drugs effective.

The Mab$^{Wt}$ load did not increase after 11 days of infection even in the dexamethasone (DEXA) -induced immunosuppression mice. A recent study showed that the *M. abscessus* burden in the lungs was unchanged after 1 week of infection using a similar DEXA-induced mouse model, but increased slightly after 2 weeks (27). This phenomenon indicates that it may need a longer period to observe a persistent uptrend in *M. abscessus* lung CFU even in immunosuppressed mice. Mab$^{\Delta embC}$ and Mab$^{\Delta CMab}$ burden of mice in the CMC-Na group at the treatment completion significantly decreased compared with that at treatment initiation, which is possibly because of their attenuated virulence ($P < 0.001$). MAB_0189 is involved in the synthesis of cell envelope, which is an important determinant of immunogenicity and pathogenicity (38). Besides, the growth defect of Mab$^{\Delta embC}$ and Mab$^{\Delta CMab}$ may provide a clue for their attenuation in mice.

Deletion of *MAB_0189c* also resulted in hypersensitivity to RFB, IMP, and LIN *in vivo*, further demonstrating the possibility of using MAB_0189 as a drug target. The contribution of virulence attenuation to the drug hypersensitive phenotype was not studied in detail, which is one limitation of the study. Mab$^{\Delta embC}$ may cause fewer or looser granulomas, thereby allowing better penetration of these antibiotics in the lung tissue, thus showing better killing effect. However, according to a recently published study, treatment duration (11 days) in this study was not enough for granuloma formation and may require ≥15 days (39), and it is widely believed that no typical granuloma could form in mouse lung. Bactericidal activities by RFB, IMP, and LIN could be observed in Mab$^{\Delta embC}$ but not for sure in Mab$^{Wt}$ and Mab$^{\Delta CMab}$ infected mice if the CFU in lungs of these groups compared with that of the corresponding CMC-Na groups (Fig. 3B). The complementary strain also showed significant attenuated virulence, which can be observed from the CMC-Na group, but RFB, IMP, and LIN did not become hypersensitive to the complementary strain *in vivo* (Fig. 3B). Therefore,

hypersensitivities to RFB, IMP, and LIN, but not to other drugs, in Mab$^{\Delta embC}$-infected mice are more likely contributed by *MAB_0189c* knockout rather than its hypovirulence. Unlike IMP, RFB, and LIN, Mab$^{\Delta embC}$ was not more susceptible to CLA, CLR and VAN *in vivo*, which indicated that the virulence attenuation cause by *MAB_0189c* knockout alone is not enough to make *M. abscessus* hypersensitive to drugs *in vivo*. Anyway, even if there is contribution of virulence attenuation to the drug hypersensitivity *in vivo*, the virulence was also led by the dysfunction of MAB_0189. In other words, the potential inhibitors of MAB_0189 may help to kill *M. abscessus* not only by enhancing the activities of multiple drugs, but also by enhancing the killing by the host through making *M. abscessus* less virulent. The inactivity of CLF is likely due to the short course of treatment (10 days). It was reported that CLF had a delayed activity against *M. abscessus*, which needs 2 weeks after treatment (40). Although CLA seemingly exhibited no activity, from the *in vivo* data, we cannot exclude that it was potentially active against Mab$^{\Delta embC}$ since its bacteriostatic activity could be possibly covered up by the downward trend of bacterial loads due to virulence attention.

An intriguing result in Mab$^{Wt}$-infected mice is that the lungs CFU of RFB and RFB + EMB treated groups were even more than that of the CMC-Na treated group ($P < 0.01$ and $P < 0.001$, respectively; Fig. 3C). This observation was confined to RFB but not to others, which infers that the phenomenon was not a generalized response to drugs. The results seem to point out a possibility that RFB triggers a mysterious mechanism that promotes *M. abscessus* growth *in vivo* or further disrupts the host immune system. A previous study showed RFB exhibits potent bactericidal activity against *M. abscessus in vivo*. The RFB MIC to *M. abscessus* used in our study is 16 $\mu$g/mL, whereas the MIC of the strain used in that study was 2.4 $\mu$M ($\approx$2.0 $\mu$g/mL) (26), which is close to the MIC (1 $\mu$g/mL) Mab$^{\Delta embC}$ here. Both studies showed the bactericidal activity of RFB (26), which indicated that *in vitro* RFB activity could well predict *in vivo* activity, though the mice were genetically modified in their study and DEXA induced in our study.

In conclusion, this study identified that *MAB_0189c* played a key role in impermeability of cell envelope, a major determinant of *M. abscessus* intrinsic antibiotic resistance and a cause of virulence attenuation. EMB can enhance the sensitivities of multiple drugs by inhibiting LAM synthesis, which might be due to inhibition of EmbC in *M. abscessus*. Taken together, potential inhibitors targeting MAB_0189 could be attractive as they might make multiple drugs inactive against *M. abscessus* effective and attenuate its virulence.

## MATERIALS AND METHODS

**Strains, cells, and culture conditions.** *Escherichia coli* DH5$\alpha$ was grown at 37°C in Luria Bertani (LB) broth and on LB agar. *M. abscessus* subsp. *abscessus* GZ002 (NCBI GenBank accession numbers CP034181), a previously described clinical isolate (41), was grown at 37°C in Middlebrook 7H9 broth (Difco) supplemented with 10% oleic acid albumin dextrose catalase (OADC, Difco) and 0.05% Tween 80, or on Middlebrook 7H10 agar (Difco) containing 10% OADC. It shows a smooth colony morphotype when grown on Middlebrook 7H11 agar. Where required, kanamycin and zeocin were used at final concentrations of 100 $\mu$g/mL and 30 $\mu$g/mL, respectively.

**Construction of Tn mutant libraries.** *M. abscessus* GZ002 transposon libraries were constructed by *Himar1* mutagenesis as described previously (42). Briefly, 100 mL of mid-log-phase *M. abscessus* culture with an optical density at 600 nm (OD$_{600}$) of $\sim$0.7 to 1.0 was incubated with $1 \times 10^{11}$ to $2 \times 10^{11}$ PFU/mL of MycoMarT7 phage at 37°C for 4 h. Subsequently, culture was washed and plated on Middlebrook 7H10 plates containing kanamycin.

**Screening and identification of Tn mutants.** Tn mutants were replica plated on Middlebrook 7H10 agar supplemented with or without a subinhibitory concentration of RIF (4 $\mu$g/mL). Colonies showing defective growth in the presence of RIF were collected and subjected to MIC measurements. The identification of Tn insertion sites in the RIF-hypersensitive clones was performed by a simple, efficient, and highly adaptable approach. Briefly, a mixture of 10 Tn mutants was prepared for the extraction of genomic DNA, followed by NGS to identify Tn insertion sites in the genomes. The reads containing transposon sequence were mapped to the genome and only insertions with the read count >20 at specific sites were considered to be true insertions. To further verify the NGS results, the disrupted region was amplified by PCR with several mixed colonies as a template in one reaction using a pair of primers targeting Tn and the disrupted gene. Several designed permutations and combinations of the mutants were used for the first batch of PCR. Usually, the insertion location of 10 mutants can be identified using 2 runs of PCR.

**Deletion and complementation of *MAB_0189c* in *M. abscessus*.** Gene deletion in *M. abscessus* was carried out as described earlier (18). Briefly, an allelic exchange substrate for generating *MAB_0189c* gene deletion was prepared by amplification of the upstream and downstream flanking arms and

subsequent cloning of them on either side of the a zeocin resistance gene and subsequently into the vector pBluescript II SK(+). The allelic exchange substrate was then amplified from the vector backbone by PCR and directly electroporated into freshly prepared electrocompetent *M. abscessus* cells carrying pJV53. Plates were incubated at 37°C for 5 days, and colonies resistant to zeocin and kanamycin were screened by PCR for identifying correct gene replacement. The marker-free deletion clones, named Mab$^{\Delta embC}$, were verified by PCR and subsequent sequencing. Complementation of the *MAB_0189c* gene was achieved by transforming pMV261-based plasmids expressing either *MAB_0189c* or *M. tuberculosis* *embC* into Mab$^{\Delta embC}$. The complemented strains were referred to as Mab$^{\Delta CMab}$ and Mab$^{\Delta CMtb}$. Primers used are listed in Table S1.

**Antibiotic susceptibility testing.** *M. abscessus* strains were grown to an OD$_{600}$ of 0.6 to 0.7. Tenfold serial dilutions were spotted on Middlebrook 7H10 plates containing RIF (2 to 8 $\mu$g/mL), VAN (2 to 8 $\mu$g/mL), CLF (0.5 to 1 $\mu$g/mL), LIN (2 to 8 $\mu$g/mL), RFB (1 to 4 $\mu$g/mL), IMP (1 to 4 $\mu$g/mL), LEV (2 to 8 $\mu$g/mL), CEF (8 to 32 $\mu$g/mL), AMK (2 to 8 $\mu$g/mL), TCG (0.125 to 0.5 $\mu$g/mL), or CLA (0.5 to 1 $\mu$g/mL). Broth dilution method was used to measure the MICs. In this assay, cells were inoculated at $5 \times 10^5$ CFU/mL into 7H9 medium with 2-fold serial drug dilutions. Cells were incubated at 37°C for 14 days for CLA and 3 days for other drugs. The MIC was defined as the lowest antibiotic concentration that prevented visible bacterial growth. The experiment was performed in triplicate and repeated twice.

**Extraction and analysis of LAM.** The *M. abscessus* strains in logarithmic growth phase (OD$_{600}$ = 0.6 to 1.0) were diluted 100 times to subculture for 24 h. EMB existed in broth for subculture at the final concentrations ($\mu$g/mL) of 0, 4, 8, and 32, respectively. Afterward, bacteria were collected by centrifugation and extracted with CHCl$_3$/CH$_3$OH (2:1) and CHCl$_3$/CH$_3$OH/H$_2$O (10:10:3). The residue was further extracted with equal volumes of water and phosphate-buffered saline (PBS)-saturated phenol at 80°C for 2 h. The aqueous layer (containing LAM and LM) was separated by SDS-PAGE (10% to 20% gradient gel) and analyzed by periodic acid/Schiff staining (20).

**Ethidium bromide uptake assay.** Ethidium bromide uptake was monitored as described earlier (43). Mab$^{Wt}$, Mab$^{\Delta embC}$, and Mab$^{\Delta CMab}$ were grown to an OD$_{600}$ of 0.6–1.0. Bacteria were collected by centrifugation and resuspended in uptake buffer (PBS, 0.05% Tween 80, pH 7.0) to an OD$_{600}$ of 0.5, followed by the addition of 25 mM glucose for energization. Ethidium bromide (1 $\mu$g/mL) was added, followed by real-time measurement in a fluorescence spectrometer (PerkinElmer) with excitation and emission wavelengths set at 520 and 595 nm, respectively. Results were normalized against fluorescence of ethidium bromide and are depicted relative to the highest fluorescence measured. The experiment was performed in triplicate and repeated three times.

**Susceptibility to chemical compounds.** As reported previously (23), stationary-phase cells were diluted in PBS to an OD$_{600}$ of 0.5, and further 10-fold serial dilutions were prepared. One microliter of each diluted solution was spotted onto 7H10 containing 0.01% (vol/vol) SDS, 2 $\mu$g/mL malachite green, and 4 $\mu$g/mL crystal violet. Cells were incubated at 37°C for 3 days.

**Mouse infection and treatment.** All animal care and experimental protocols were approved by the Committee on Laboratory Animal Ethics of Guangzhou Institutes of Biomedicine and Health (GIBH), Chinese Academy of Sciences. Five- to six-week-old, female BALB/c mice (Charles River) were infected with 10 mL *M. abscessus* culture at an OD$_{600}$ of 0.7–0.8 via aerosolization using a Glas-Col inhalation exposure system (Glas-Col, Terre Haute, Indiana) with inhalation time 45 min according to the manufacturer's instructions.

To achieve adequate immunosuppression, mice were treated with DEXA. DEXA (D1756, Sigma-Aldrich) was dissolved in sterile 1× phosphate-buffered saline (PBS, pH 7.4) and administered via daily subcutaneous injection at 5 mg/kg/day, as described previously (27). Daily DEXA treatment began 2 weeks prior to infection and was continued throughout the duration of the experiment. Mice were sacrificed at 24 h postinfection to determine the initial bacterial burden in the lungs.

**Antibiotics and regimens.** The mice were treated at doses (mg/kg) via oral gavage once daily with 0.4% CMC-Na as the solvent, CLR 250, CLF 50, LIN 100, RFB 20, and EMB 200, or via subcutaneous injection twice daily with either VAN 110 or IMP 100. Combination treatment groups were as follows: EMB + IMP, EMB + CLF, EMB + RFB, and EMB + CLA. Mice were sacrificed for CFU determination in lungs at 11 days after treatment initiation. Drug stocks (mg/mL) were dissolved in sterile 0.4% CMC-Na: CLR 25, CLF 5, LIN 10, RFB 2, EMB 20, or in 1× PBS: VAN, 11, IMP 10. Additionally, cilastatin was added to IMP preparation in a 1:1 dose ratio aimed to compensate for higher activity of murine renal dihydropeptidase-1 compared to humans, as described previously (44).

## SUPPLEMENTAL MATERIAL

Supplemental material is available online only.
**SUPPLEMENTAL FILE 1**, PDF file, 0.5 MB.

## ACKNOWLEDGMENTS

This work was supported by the National Key R&D Program of China (2021YFA1300904), partially by the National Natural Science Foundation of China (NSFC 81973372), Joint Research the Russian Science Foundation (RSF)-NSFC Collaboration grant numbers 21-45-00018 (to D.M.), 82061138019 (to T.Z.), the Joint Research Health Research Council of New Zealand (HRC)-NSFC Collaboration grant (82061128001), Department of Science and Technology of Guangdong Province (2019B110233003), the Chinese Academy of Sciences

(154144KYSB20190005, YJKYYQ20210026), China-New Zealand Joint Laboratory on Biomedicine and Health and China Postdoctoral Science Foundation (2021M701575). The funders had no role in study design, data collection, and analysis, decision to publish, or preparation of the manuscript.

All authors read and approved the final version of the manuscript.

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
