## [Reviewer comments · Microbiology Spectrum]

Microbiology Spectrum

Arabinosyltransferase C Mediates Multiple Drugs Intrinsic Resistance by Altering Cell Envelope Permeability in *Mycobacterium abscessus*

Shuai Wang, Xiaoyin Cai, Wei Yu, Sheng Zeng, Jingran Zhang, Lingmin Guo, Yamin Gao, Zhili Lu, H.M. Adnan Hameed, Cuiting Fang, Xirong Tian, Buhari Yusuf, Chiranjibi Chhotaray, Md Alam, Buchang Zhang, Hong-Hua Ge, Dmitry A. Maslov, Gregory M. Cook, Jiacong Peng, Yong-Ping Lin, Nanshan Zhong, Guoliang Zhang, and Tianyu Zhang

Corresponding Author(s): Tianyu Zhang, Guangzhou Institutes of Biomedicine and Health, Chinese Academy of Sciences; University of Chinese Academy of Sciences

Review Timeline:

Submission Date:	December 28, 2021
Editorial Decision:	February 11, 2022
Revision Received:	April 12, 2022
Editorial Decision:	May 2, 2022
Revision Received:	June 21, 2022
Accepted:	July 18, 2022

Editor: Digby Warner

Reviewer(s): The reviewers have opted to remain anonymous.

Transaction Report:

DOI: <https://doi.org/10.1128/spectrum.02763-21>

February 11, 2022

Prof. Tianyu Zhang

Guangzhou Institutes of Biomedicine and Health, Chinese Academy of Sciences; University of Chinese Academy of Sciences
State Key Laboratory of Respiratory Disease
Room 207, 190 Kaiyuan Rd
Science Park
Guangzhou, Guangdong 510530
China

Re: Spectrum02763-21 (Arabinosyltransferase C Inhibitors may Enhance Activities of Multiple Drugs by Altering Cell Wall Permeability in *Mycobacterium abscessus*)

Dear Prof. Tianyu Zhang:

Thank you for submitting your manuscript to Microbiology Spectrum.

As detailed below, the reviewers were in agreement that the results presented in this manuscript are interesting but do not provide adequate support for the major claims. Specific concerns which must be addressed include:

1. The design and analysis of the transposon screen: this screen, which provides the basis for the study, must be more clearly described and the full results thereof explained - otherwise it undermines the credibility of the subsequent analyses of MAB_0189c (embC).
2. The interpretation that inhibition of EmbC might potentiate activity of other antimycobacterial agents is intriguing but the evidence supporting this possibility must be strengthened considerably.
3. The impact of the embC deletion on *M. abscessus* fitness and pathogenicity might obscure interpretation of the in vivo results; this must be considered.
4. Throughout the manuscript, lack of clarity and/or missing details make it difficult to follow the narrative - please ensure that all necessary revisions are made to increase the readability of the work.

Link Not Available

Sincerely,

Digby Warner

Journals Department
Reviewer comments:

Reviewer #1 (Comments for the Author):

The manuscript by Wang et al. reports the MAB_0189c (embC) gene, encoding the arabinosyltransferase involved in polymerization of the arabinan portion of lipoarabinomannan (LAM) as a key factor in the intrinsic resistance of *M. abscessus* to several classes of antibiotics. This gene was identified by screening a *M. abscessus* transposon library and selection of hypersensitivity to drugs. Complementation of the mutant partially restored the WT phenotypes. Disruption of MAB_0189c was associated a reduced production of LAM and enhanced cell wall permeability. In addition, exposure to subinhibitory concentrations of ethambutol (EMB), a known drug targeting the arabinosyltransferases, including EmbC, sensitized *M. abscessus* to several antibiotics in vitro but not in a mouse model of infection, which somehow limits the impact of the study. Although the aim of this work is of importance to the mycobacterial community and the results help at better understanding the intrinsic resistance mechanisms occurring in *M. abscessus*, the study requires additional controls to strengthen the data. Moreover, some the conclusions raise are not always supported by strong data. At many instances, the description of the experiments (particularly the figure legends) are difficult to follow and require more details.

Major concerns:

- The authors mention having identified 10 individual mutants hypersensitive to RIF in their screen. However, they only focused on MabZ6. What about the other mutants ? Are they also mutated in MAB_0189c or are they affected in other genes? The authors should present a Table with the different genes identified in their screen to highlight whether or not MAB_0189c is a primary gene involved in resistance to antibiotics. Why did they not further characterize the remaining mutants ?
- Figure 1 is not complete. The activity of tigecycline and amikacine on the different strains should be shown, as mentioned in line 142.
- Figure 2. The method used to identify the Tn insertion is not well described in the text and the figure legend. Difficult to understand for non-experts in the field.
- Figure S2. The mutant has a clear growth defect in vitro. Has this been observed in other media or is it specific to 7H9? This growth defect may explain the reduced virulence phenotype found in mice. This should be better described in the text (Discussion section).
- Analysis of the LM/LAM profile by SDS-PAGE indicates reduced levels of LAM as compared to LM in the mutant, presumably explaining the increase permeability of the cell wall. This prompted the authors to test whether ethambutol (EMB) know to inhibit the arabinosyltransferases, including EmbC, could sensitize *M. abscessus* to multiple antibiotics. However, they need to show that at the concentrations tested, EMB affects the synthesis of LAM in *M. abscessus* (reduced levels are expected) by providing the LAM/LM profile of WT cultures treated with EMB. This is important to support their hypothesis and claims.
- The presentation of the manuscript could be greatly improved by combining some figures to make it easier for the reader to follow the study. For instance, Fig 3 should be combined with Fig 2 as it presents the results of Tn insertion. Fig 6 should be combined with Fig 5 as it relates studies in mice.
- Fig 5. in the absence of a schema illustrating the protocol used, it is difficult to understand how the experiments was performed. More details are also required in the figure legend.

Minor points:

- what is the morphotype of GZ002? This is important to report since the presence (in S) or absence (in R) of the GPL outer layer plays also a role in the permeability of the cell wall.
- line 173 "resistance"
- Figure legend 5, line 714. "EMB 200". EMB has not been used in this experiment.
- Table 2. Do these values correspond to MICs ?
- Figure S2. The legend is incomplete. What are the expected sizes of the various amplicons ?
- Figure S3 is useless. If the authors want to keep EM pictures, they should provide images at higher magnification to see eventual differences in the cell wall layers in the different strains.

Reviewer #2 (Comments for the Author):

Wang et al. report on a *Mycobacterium abscessus* MAB_0189c (embC) knockout mutant that displays increased cell envelope permeability resulting in hypersusceptibility to a number of antibiotics in vitro and, potentially, in vivo. The embC mutant was originally identified by screening a library of *M. abscessus* transposon mutants for clones with increased susceptibility to rifampicin. A clean deletion mutant was subsequently generated by homologous recombination to confirm these phenotypes.

While the finding that the embC gene of *M. abscessus* is not essential is by itself interesting, it is not entirely surprising since embC has been reported to be a non-essential gene in all rapidly-growing nontuberculous mycobacteria that have been examined to date (*M. smegmatis*, *M. neoaurum*). Disruption of embC in these other species was also shown to enhance

antibiotic susceptibility. This somewhat diminishing the novelty of the results presented herein.

More interesting, perhaps, is the hypothesis that specific inhibitors of EmbC (to be identified) may potentiate the activity of a variety of antibiotics used in combination *in vivo*. While this may be true, the *in vivo* data presented in Fig. 5 do not conclusively prove this assumption since the outcome of treating pharmacologically an established *M. abscessus* murine infection with an EmbC inhibitor may be very different from treating mice infected with the embC knock-out mutant which is totally deficient in EmbC activity from the onset of infection and possibly attenuated for virulence/pathogenicity. Hence, an alternative explanation for the enhanced activity of IMP, LIN and RFB against the mutant may be that the mutant causes fewer granulomatous lung lesions thereby allowing better penetration of these antibiotics in the lung tissue and, thus, better killing. This possible interpretation of the results is not discussed, nor is the pathogenicity of the embC knock-out vs WT strain examined. Therefore, whether EmbC inhibitors would potentiate the activity of other antibiotics remains an open question that may only be addressed either through the use of gene silencing or that of EmbC inhibitors as they become available.

The use of ethambutol to prove the point is not helpful here because (i) EMB actually does not show any synergism with other antibiotics *in vivo* (Fig. 6); and (ii) no evidence is provided in the manuscript that EMB actually inhibits EmbC in *M. abscessus* (despite what is said in the text, e.g., p. 11, lines 211-212). Have the authors examined the effect of EMB on *M. abscessus* LAM and arabinogalactan synthesis? This is particularly intriguing because *M. abscessus* is normally highly resistant to EMB because of a naturally occurring mutation in the embB gene which is involved in arabinogalactan synthesis (missing reference in the text). Do the isolates used in this study exhibit this embB SNP?

Another concern is with the acute, immunosuppressed BALB/c mouse model used in Figures 5 and 6. It is unclear how the infection progresses in this model since a control group that did not receive any treatment (not even the CMC-Na vehicle) is missing. The WT bacterial load does not increase after 10 days which is surprising given the DEXA-induced immunosuppression. The mutant CFU and, surprisingly, the complemented mutant CFU, significantly decrease. The authors hypothesize this is because of the virulence attenuation of the mutant and inefficient complementation in the complemented strain *in vivo*. Could the vehicle itself have caused a decrease in bacterial load? This needs to be established for proper interpretation of the results. It is also very surprising that RBT shows no activity against the WT strain in this model since half the dose of RFB as used herein has been established previously to display potent bactericidal activity against *M. abscessus* *in vivo* (cf. studies by the Dick's laboratory).

The SDS-PAGE of LAM extracted from the different strains presented in Fig. 4 is very dark and LAM is barely visible. LM and LAM from *M. abscessus* should present as two distinct bands on silver-stained SDS-PAGE. At present, it is impossible to tell if LAM synthesis is deficient in the KO and is restored in the complemented mutant. Lipoglycan extraction needs to be optimized and the presence/absence of LAM in the different strains confirmed by immunoblot.

The Results section on "Increased cell wall permeability" is not very well written. Line 189-190, what does "disruption of the arabinan layer" mean? There seems to be some confusion between the respective impact of LAM (embC) and arabinogalactan (embA and embB) on cell envelope permeability. The two are not interrelated.

Other comments

The discussion is unnecessarily long and could be significantly shortened.

There seems to be some confusion throughout the text between the terms "cell wall" and "cell envelope". LAM is not a cell wall component as it is not covalently attached to any cell wall core component.

Two-fold difference in MIC in tables 1 and 2 are probably within the margin of experimental error and are usually not considered significant.

The method used to map the transposon insertion sites in a pool of mutants is also not all that innovative.

Figures 2 and 3 could be moved to the Supplementary Materials.

Line 194: "complementation", not "expression".

What *M. abscessus* subspecies did the authors use in this study?

The resolution of Fig. S3 is rather poor.

Staff Comments:

Preparing Revision Guidelines

Please return the manuscript within 60 days; if you cannot complete the modification within this time period, please contact me. If you do not wish to modify the manuscript and prefer to submit it to another journal, please notify me of your decision immediately so that the manuscript may be formally withdrawn from consideration by Microbiology Spectrum.

Title: Arabinosyltransferase C Inhibitors may Enhance Activities of Multiple Drugs by Altering Cell Wall Permeability in *Mycobacterium abscessus*

ID: Spectrum02763-21

11th April, 2022

Dear Editors and Reviewers:

Thank you for your letter and for the reviewers' comments concerning our manuscript entitled "Arabinosyltransferase C Inhibitors may Enhance Activities of Multiple Drugs by Altering Cell Wall Permeability in *Mycobacterium abscessus*" (ID: Spectrum02763-21). These comments are all valuable and very helpful for us to revise and improve the paper. We have considered all the comments carefully and made corrections accordingly and we hope that the modified manuscript can be finally accepted.

The main corrections are marked in red in the revised paper. Responses to the reviewer's comments are as follows:

Reviewer #1 (Comments for the Author):

The manuscript by Wang et al. reports the *MAB_0189c* (*embC*) gene, encoding the arabinosyltransferase involved in polymerization of the arabinan portion of lipoarabinomannan (LAM) as a key factor in the intrinsic resistance of *M. abscessus* to several classes of antibiotics. This gene was identified by screening a *M. abscessus* transposon library and selection of hypersensitivity to drugs. Complementation of the mutant partially restored the WT phenotypes. Disruption of *MAB_0189c* was associated a reduced production of LAM and enhanced cell wall permeability. In addition, exposure to subinhibitory concentrations of ethambutol (EMB), a known drug targeting the arabinosyltransferases, including EmbC, sensitized *M. abscessus* to several antibiotics in vitro but not in a mouse model of infection, which somehow limits the impact of the study.

Although the aim of this work is of importance to the mycobacterial community and the results help at better understanding the intrinsic resistance mechanisms occurring in

M. abscessus, the study requires additional controls to strengthen the data. Moreover, some the conclusions raise are not always supported by strong data. At many instances, the description of the experiments (particularly the figure legends) are difficult to follow and require more details.

Major concerns:

1. The authors mention having identified 10 individual mutants hypersensitive to RIF in their screen. However, they only focused on Mab^{Z6}. What about the other mutants? Are they also mutated in *MAB_0189c* or are they affected in other genes? The authors should present a Table with the different genes identified in their screen to highlight whether or not *MAB_0189c* is a primary gene involved in resistance to antibiotics. Why did they not further characterize the remaining mutants?

Authors' response:

First of all we are thankful to the reviewer for your clear understanding and describing the precise theme of our study.

Regarding your comment, we acknowledge your concern. Therefore, we supplemented the information of the remaining transposon mutants in table S1 in the 'supplementary material' of revised manuscript. Here we want to clarify that the functions of the genes disrupted by transposon belong to different categories, so we could not study all of them together in one study, where our aim was to mainly investigate the key role of *embC* in intrinsic resistance in *M. abscessus*. However, the remaining genes will be comprehensively investigated in our future studies.

2. Figure 1 is not complete. The activity of tigecycline and amikacin on the different strains should be shown, as mentioned in line 142.

Authors' response: Thanks for your suggestion. We have modified the Figure 1 which contains the activity of amikacin and tigecycline on the different strains. We also described in the revised manuscript(Lines 142-145).

3. Figure 2. The method used to identify the Tn insertion is not well described in the text and the figure legend. Difficult to understand for non-experts in the field.

Authors' response: Thanks for your suggestion. Figure 2 is now moved to the supplementary material of the revised manuscript which is now marked as 'Figure S1'. We have modified the figure and its legend.

4. Figure S2. The mutant has a clear growth defect *in vitro*. Has this 'been observed in other media or is it specific to 7H9? This growth defect may explain the reduced virulence phenotype found in mice. This should be better described in the text (Discussion section).

Authors' response: Thanks for your suggestion. We detected the growth of wild-type *M. abscessus*, knockout strains and complementary strains only in 7H9 medium. We have described it in the discussion (Lines 325-327) .

5. Analysis of the LM/LAM profile by SDS-PAGE indicates reduced levels of LAM as compared to LM in the mutant, presumably explaining the increase permeability of the cell wall. This prompted the authors to test whether ethambutol (EMB) know to inhibit the arabinosyltransferases, including EmbC, could sensitize *M. abscessus* to multiple antibiotics. However, they need to show that at the concentrations tested, EMB affects the synthesis of LAM in *M. abscessus* (reduced levels are expected) by providing the LAM/LM profile of WT cultures treated with EMB. This is important to support their hypothesis and claims.

Authors' response: We gratefully appreciate your valuable suggestion. We have detected the effect of EMB inhibition on LAM biosynthesis in *M. abscessus* in our revised manuscript. We did not find significant changes in the content of LAM at 4 µg/mL and 8 µg/mL of EMB. We speculate that this may be because EMB cannot inhibit the growth of *M. abscessus*, so these concentrations of EMB are not enough to effectively inhibit the synthesis of LAM (Figure 2A: lane 4-5). However, when we increased the concentration of EMB to 32 µg/mL, we observed the clear inhibition of LAM synthesis (Figure 2A: lane 6), indicating that EmbC activity was inhibited directly.

6. The presentation of the manuscript could be greatly improved by combining some figures to make it easier for the reader to follow the study. For instance, Fig 3 should be combined with Fig 2 as it presents the results of Tn insertion. Fig 6 should be combined with Fig 5 as it relates studies in mice.

Authors' response: Thanks for your suggestion. We have combined the 'Figure 2 and Figure 3' to merge them into one figure which is placed now in the supplementary material and named it as 'Figure S1'. Similarly, the 'Figure 5 and Figure 6' are combined and named it as 'Figure 3' in the revised manuscript.

7. Fig 5. in the absence of a schema illustrating the protocol used, it is difficult to understand how the experiments was performed. More details are also required in the figure legend.

Authors' response: We gratefully acknowledge your concern. Following your recommendation, we have added a schematic diagram of animal experiments in the revised manuscript (Figure 3A). The figure legends are modified in the revised manuscript as well.

8. Minor points:

- i. what is the morphotype of GZ002? This is important to report since the presence (in S) or absence (in R) of the GPL outer layer plays also a role in the permeability of the cell wall.

Authors' response: We used *M. abscessus* GZ002, which is smooth type and a subspecies of abscessus, which is described in the revised manuscript (Lines 394-395).

- ii. line 173 "resirance"

Authors' response: The 'resirance' has been modified to 'resistance' in the revised manuscript (Line 179).

- iii. Figure legend 5, line 714. "EMB 200". EMB has not been used in this experiment.

Authors' response: The 'Figure 5 and Figure 6' are combined and named as 'Figure

3' in the revised manuscript and updated the figure legend accordingly.

iv. Table 2. Do these values correspond to MICs?

Authors' response: The Table 2 corresponds to the drug concentration values of the antibiotics influenced by the combined application of EMB concentrations and the tested drugs.

v. Figure S2. The legend is incomplete. What are the expected sizes of the various amplicons ?

Authors' response: We have modified the 'Figure S2'. The sizes of the amplicons are added and updated the figure legend.

vi. Figure S3 is useless. If the authors want to keep EM pictures, they should provide images at higher magnification to see eventual differences in the cell wall layers in the different strains.

Authors' response: We have deleted the 'Figure S3' from the revised manuscript.

Thanks again for all of your valuable suggestions.

Reviewer #2 (Comments for the Author):

Wang et al. report on a *Mycobacterium abscessus* MAB_0189c (*embC*) knockout mutant that displays increased cell envelope permeability resulting in hypersusceptibility to a number of antibiotics *in vitro* and, potentially, *in vivo*. The *embC* mutant was originally identified by screening a library of *M. abscessus* transposon mutants for clones with increased susceptibility to rifampicin. A clean deletion mutant was subsequently generated by homologous recombination to confirm these phenotypes.

While the finding that the *embC* gene of *M. abscessus* is not essential is by itself interesting, it is not entirely surprising since *embC* has been reported to be a non-essential gene in all rapidly-growing nontuberculous mycobacteria that have been examined to date (*M. smegmatis*, *M. neoaurum*). Disruption of *embC* in these other species was also shown to enhance antibiotic susceptibility. This somewhat diminishing the novelty of the results presented herein.

1. More interesting, perhaps, is the hypothesis that specific inhibitors of EmbC (to be identified) may potentiate the activity of a variety of antibiotics used in combination *in vivo*. While this may be true, the *in vivo* data presented in Fig. 5 do not conclusively prove this assumption since the outcome of treating pharmacologically an established *M. abscessus* murine infection with an EmbC inhibitor may be very different from treating mice infected with the *embC* knock-out mutant which is totally deficient in EmbC activity from the onset of infection and possibly attenuated for virulence/pathogenicity. Hence, an alternative explanation for the enhanced activity of IMP, LIN and RFB against the mutant may be that the mutant causes fewer granulomatous lung lesions thereby allowing better penetration of these antibiotics in the lung tissue and, thus, better killing. This possible interpretation of the results is not discussed, nor is the pathogenicity of the *embC* knock-out vs WT strain examined. Therefore, whether EmbC inhibitors would potentiate the activity of other antibiotics remains an open question that may only be addressed either through the use of gene silencing or that of EmbC inhibitors as they become available.

Authors' response: First of all, we are thankful to you for precisely describing the theme of our study.

We partially agree with the reviewer's explanation that the decrease in bacterial load in the lungs of mice infected with the knockout strain and treated with IMP, LIN and RFB may be caused by the inability to form granuloma due to the reduced virulence of the knockout strain. Similarly, another possibility of this decrease in bacterial load is because of retaining the short duration (10 days) of infection in our study, whereas, according to some other recently published studies this duration is not enough for granuloma formation which may require ≥ 15 days (1). We have elaborated this in the 'discussion' of the revised manuscript (Lines 329-336). We are also screening some other effective *M. abscessus* EmbC inhibitors for our future studies.

2. The use of ethambutol to prove the point is not helpful here because (i) EMB actually does not show any synergism with other antibiotics *in vivo* (Fig. 6); and (ii) no evidence is provided in the manuscript that EMB actually inhibits EmbC in *M. abscessus* (despite what is said in the text, e.g., p. 11, lines 211-212). Have the authors examined the effect of EMB on *M. abscessus* LAM and arabinogalactan synthesis? This is particularly intriguing because *M. abscessus* is normally highly resistant to EMB because of a naturally occurring mutation in the *embB* gene which is involved in arabinogalactan synthesis (missing reference in the text). Do the isolates used in this study exhibit this *embB* SNP?

Authors' response: Thanks for your comments. (1) We acknowledge your concern, in fact, the absence of synergistic activity of EMB with other antibiotics *in vivo* is because the EMB cannot inhibit the growth of *M. abscessus* as it is intrinsically resistant to EMB. It has also been observed that the low concentrations of EMB cannot completely inhibit the synthesis of LAM *in vitro*, so, we speculate that the similar phenomena exists *in vivo*. Therefore, the combination of EMB and other drugs could not achieve the effect of knockout of *embC*.

(2) We have detected the effect of EMB inhibition on LAM biosynthesis in *M.*

abscessus in our revised manuscript. We did not find significant changes in the content of LAM at 4 µg/mL and 8 µg/mL of EMB. We propose that this is possibly due to the reason that the growth of *M. abscessus* cannot be inhibited by low concentration of EMB, as it was shown in (Figure 2A: lane 4-5) that 4-8 µg/mL of EMB is not enough to completely inhibit the synthesis of LAM. However, when we increased the concentration of EMB to 32 µg/mL, we observed the clear inhibition of LAM synthesis (Figure 2A: lane 6), indicating that EmbC activity was inhibited directly. EmbCs from *M. abscessus* and *M. tuberculosis* are different, so EmbC may not inhibit EmbC efficiently. If better inhibitor(s) targeting EmbC of *M. abscessus* can be discovered, it may reverse its intrinsic resistance to many drugs.

(3) In addition, we also compared the EmbB amino acid sequences of *M. abscessus* and *Mycobacterium tuberculosis* in the revised manuscript. We found that *M. abscessus* GZ002 exhibit the similar *embB* amino acid substitution. The I281Q substitution involves in EMB resistance in *M. abscessus* and designated as the EMB resistance-determining region (Fig. S4). We have also described this in the discussion section (Lines 297-232).

3. Another concern is with the acute, immunosuppressed BALB/c mouse model used in Figures 5 and 6. It is unclear how the infection progresses in this model since a control group that did not receive any treatment (not even the CMC-Na vehicle) is missing. The WT bacterial load does not increase after 10 days which is surprising given the DEXA-induced immunosuppression. The mutant CFU and, surprisingly, the complemented mutant CFU, significantly decrease. The authors hypothesize this is because of the virulence attenuation of the mutant and inefficient complementation in the complemented strain *in vivo*. Could the vehicle itself have caused a decrease in bacterial load? This needs to be established for proper interpretation of the results. It is also very surprising that RBT shows no activity against the WT strain in this model since half the dose of RFB as used herein has been established previously to display potent bactericidal activity against *M. abscessus in vivo* (cf. studies by the Dick's laboratory).

Authors' response: We acknowledge your concern and respectfully want to clarify that in this study we used CMC-Na as a vehicle in animal experiments (Figure 3 B, C red columns). We found that there was no any increase in the mean burden in the lungs of mice infected with wild type *M. abscessus* after infection day 1 to day 11. This result is similar to the previous study where no significant increase was reported in lungs bacterial load from the first day to the 11th day after infection with *M. abscessus* K21 in NOD SCID mice (2). Through the drug sensitivity results of the growth curve, we observed that the complementary strains were not completely reverted to the wild-type phenotype. The growth of complementary strains was still slower than that of wild-type strains. This growth defect of complementary and knockout strains may explain the decreased CFU counts in mouse lung. We have also described this in detail in the 'discussion' section of the revised manuscript (Lines 317-320 and 325-327).

Moreover, the clinical *M. abscessus* strains may have the different sensitivities to RFB (3, 4). The MIC of RFB to *M. abscessus* used in our study is 16 µg/mL, whereas the MIC of the strain used in another study was 2.4 µM (≈ 2.0 µg/mL) (2). This can be the possible reason why RFB showed no activity against the wild type strain *in vivo* in our study. We have explained this in the 'discussion' of the revised manuscript (Lines 350-355).

4. The SDS-PAGE of LAM extracted from the different strains presented in Fig. 4 is very dark and LAM is barely visible. LM and LAM from *M. abscessus* should present as two distinct bands on silver-stained SDS-PAGE. At present, it is impossible to tell if LAM synthesis is deficient in the KO and is restored in the complemented mutant. Lipoglycan extraction needs to be optimized and the presence/absence of LAM in the different strains confirmed by immunoblot.

Authors' response: We appreciate your suggestion. To solve the concerned issue, we reformed the relevant experiments and LM and LAM were extracted again to proceed this experiment. The LM and LAM bands from *M. abscessus* appeared as distinct bands on silver-stained SDS-PAGE in the new 'Figure 2A' of the revised

manuscript. We also compared the LM and LAM of *M. abscessus* with that of a recently published study, and their sizes were relatively similar (5), so we did not carry out immunoblot.

5. The results section on "Increased cell wall permeability" is not very well written. Line 189-190, what does "disruption of the arabinan layer" mean? There seems to be some confusion between the respective impact of LAM (*embC*) and arabinogalactan (*embA* and *embB*) on cell envelope permeability. The two are not interrelated.

Authors' response: Thank you for pointing out this problem in manuscript. We have modified the concerned text in the revised manuscript.

6. Other comments

i. The discussion is unnecessarily long and could be significantly shortened.

Authors' response: Following your recommendation, we have shortened the 'discussion' section.

ii. There seems to be some confusion throughout the text between the terms "cell wall" and "cell envelope". LAM is not a cell wall component as it is not covalently attached to any cell wall core component.

Authors' response: We have thoroughly checked and corrected the terms "cell wall" and "cell envelope".

iii. Two-fold difference in MIC in tables 1 and 2 are probably within the margin of experimental error and are usually not considered significant.

Authors' response: For MIC testing the experiment was performed in triplicate and repeated twice which were later confirmed by different authors of this research team. Every time our analysis showed the similar results which considerably countered the chances of experimental error and endorsed our outcome.

iv. The method used to map the transposon insertion sites in a pool of mutants is also not all that innovative.

Authors' response: Although, Tn-seq is not a new method, to the best of our knowledge it is the first time to use the whole genome sequencing combined with PCR to identify the transposon insertion site in a single transposon mutant. The method used in our study is simple and highly adaptable compared to the conventional methods which can identify both accurately and effectively ≥ 10 Tn insertions sites using only one NGS in combination with multiplex PCR. Besides its fast performance, this method is also more efficient and cost-effective tool for identifying the particular Tn insertion sites.

v. Figures 2 and 3 could be moved to the Supplementary Materials.

Authors' response: In the revised manuscript, we have combined the 'Figure 2 and Figure 3' into one figure and named it as 'Figure S1' which is placed now in the supplementary material.

vi. Line 194: "complementation", not "expression".

Authors' response: We have changed "expression" to "complementation" in the revised manuscript (Line 199).

vii. What *M. abscessus* subspecies did the authors use in this study?

Authors' response: In the revised manuscript, we have mentioned the subspecies of *M. abscessus* in line 390 as '*M. abscessus* subsp. *abscessus*'.

viii. The resolution of Fig. S3 is rather poor.

Authors' response: There was no significant difference in the cell wall structure of these strains and suggested by other reviewers as well we have deleted the 'Fig. S3' in our revised manuscript.

Thanks again for all of your valuable suggestions.

References

1. Rottman M, Catherinot E, Hochedez P, Emile JF, Casanova JL, Gaillard JL, Soudais C. 2007. Importance of T cells, gamma interferon, and tumor necrosis factor in immune control of the rapid grower *Mycobacterium abscessus* in C57BL/6 mice. *Infect Immun* 75:5898-907.

2. Dick T, Shin SJ, Koh WJ, Dartois V, Gengenbacher M. 2020. Rifabutin is active against *Mycobacterium abscessus* in mice. *Antimicrob Agents Chemother* 64:e01943-19.
3. Johansen MD, Daher W, Roquet Banères F, Raynaud C, Alcaraz M, Maurer FP, Kremer L. 2020. Rifabutin is bactericidal against intracellular and extracellular forms of *Mycobacterium abscessus*. *Antimicrob Agents Chemother* 64: e00363-20.
4. Aziz DB, Low JL, Wu ML, Gengenbacher M, Teo JWP, Dartois V, Dick T. 2017. Rifabutin is active against *Mycobacterium abscessus* complex. *Antimicrob Agents Chemother* 61: e00155-17.
5. Palcekova Z, Gilleron M, Angala SK, Belardinelli JM, Jackson MC. 2020. Polysaccharide succinylation enhances the intracellular survival of *Mycobacterium abscessus*. *ACS Infect Dis* 6: 2235-2248.

Yours sincerely,

Tianyu Zhang, Ph.D, Professor,

State Key Laboratory of Respiratory Disease,

Guangdong-Hong Kong-Macao Joint Laboratory of Respiratory Infectious Disease,

Guangzhou Institutes of Biomedicine and Health (GIBH),

Chinese Academy of Sciences (CAS).

May 2, 2022

Prof. Tianyu Zhang

Guangzhou Institutes of Biomedicine and Health, Chinese Academy of Sciences; University of Chinese Academy of Sciences
State Key Laboratory of Respiratory Disease
Room 207, 190 Kaiyuan Rd
Science Park
Guangzhou, Guangdong 510530
China

Re: Spectrum02763-21R1 (Arabinosyltransferase C Inhibitors may Enhance Activities of Multiple Drugs by Altering Cell Wall Permeability in *Mycobacterium abscessus*)

Dear Prof. Tianyu Zhang:

Thank you for submitting your revised manuscript to Microbiology Spectrum.

As detailed in the reviewer comments below, there is general appreciation that the revisions have strengthened this submission. However, there are persistent concerns which must be addressed before this work can be considered suitable for publication, specifically:

- (i) key experimental details are missing which must be included;
- (ii) the mouse model must be described in greater detail and the in vivo results interpreted accordingly;
- (iii) the Discussion section is overlong and can be shortened significantly by limiting the interpretation of the results to that which is reasonably supported by the experimental observations.

Link Not Available

Sincerely,

Digby Warner

Journals Department
Reviewer comments:

Reviewer #1 (Comments for the Author):

The authors have satisfactorily responded to all my previous concerns. The conclusions raised are now strengthened by additional data and appropriate controls.

There are still a few typos that need to be corrected:

-line 173. I suppose the authors refer here to Figure S3 and not S2 (in vitro growth curve)

-line 301: "glutarnine"

In addition, the Discussion is still very long (6 pages) and could certainly be shortened.

Reviewer #2 (Comments for the Author):

The revision only very partially addresses my concerns and the writing of the manuscript still requires significant improvement. The LAM extraction/gel separation (now presented as Fig. 2A) has been improved, providing more convincing evidence for the LAM-deficient phenotype of the embC knock-outs and partial restoration in the complemented strain. Treatment of *M. abscessus* with ethambutol at high concentrations (32 mg/L) leads to inhibition of LAM synthesis supporting the inhibition of EmbC by EMB in this *Mycobacterium* species, albeit at concentrations higher than those used in the in vitro drug synergism study (Table 2). However, how this experiment was performed in neither described in the legend of figure 2 or in the Materials and Methods. When was EMB added? For how long? These are critical details that need to be added.

EMB shows no synergism with other antibiotics in vivo (Fig. 3C). The embC mutant is more susceptible to a number of drugs in 7H9 medium and in vivo (Table 1 and Fig. 3A) but to what extent the hypovirulence of the mutant contributes to this phenotype in vivo remains unclear. No clear pathogenicity data (including pathology and AFB staining) are shown. Again, the outcome of treating pharmacologically an established *M. abscessus* murine infection with an EmbC inhibitor may be very different from treating mice infected with the embC knock-out mutant which is totally deficient in EmbC activity from the onset of infection and attenuated for virulence and pathogenicity. Therefore, the conclusions of the study remain rather speculative and do not significantly add to the field beyond what was already known of the hypersusceptibility of embC mutants to antibiotics in rapidly growing mycobacteria.

The discussion was not shortened. It is actually longer than in the previous version of the manuscript. It is still unfocused and not clearly written with contradictory statements on the impact of pathogenicity on drug susceptibility (p. 17, lines 329-336) and in other places as well (e.g., lines 267-269).

The murine model used in the study remains poorly characterized. A reference is now cited to justify the lack of increase in bacterial burden between day 1 and day 10 but it's completely unrelated to the dexamethazone treatment model used here and, therefore, doesn't add anything to the discussion.

Other concerns:

p. 10, line 172: Fig. S3 (not S2)

p. 10, line 183: complementation (not "completion")

p. 11, lines 209-210: What is a "nightmare" bacterium? Delete "nightmare"

p. 12: the reason for the lack of killing activity of EMB against *M. abscessus* should be explained in light of the sequencing of embB in the same isolate.

13, lines 251-253: EMB may not penetrate inside *M. abscessus* enough to reach its target.

p. 15, lines 278-279: "disruption" of what exactly?

Fig. 1: Typo (Tigecycline)

Staff Comments:

Preparing Revision Guidelines

- Point-by-point responses to the issues raised by the reviewers in a file named "Response to Reviewers," NOT IN YOUR COVER LETTER.
- Upload a compare copy of the manuscript (without figures) as a "Marked-Up Manuscript" file.
- Each figure must be uploaded as a separate file, and any multipanel figures must be assembled into one file.

- Manuscript: A .DOC version of the revised manuscript
- Figures: Editable, high-resolution, individual figure files are required at revision, TIFF or EPS files are preferred

Please return the manuscript within 60 days; if you cannot complete the modification within this time period, please contact me. If you do not wish to modify the manuscript and prefer to submit it to another journal, please notify me of your decision immediately so that the manuscript may be formally withdrawn from consideration by Microbiology Spectrum.

Title: Arabinosyltransferase C Mediates Multiple Drugs Intrinsic Resistance by Altering Cell Envelope Permeability in *Mycobacterium abscessus*

ID: Spectrum02763-21

21th June, 2022

Dear Editors and Reviewers:

On behalf of my co-authors, we are thankful to you for providing us an opportunity to revise our manuscript, we highly appreciate the reviewers for their constructive suggestions to improve our manuscript, entitled “Arabinosyltransferase C Mediates Multiple Drugs Intrinsic Resistance by Altering Cell Envelope Permeability in *Mycobacterium abscessus*” (Manuscript ID: Spectrum02763-21). We have read the comments carefully and made revision accordingly. The main revisions in the manuscript and the response to the reviewers’ comments are elucidated in detail in the attached file “Responses to comments”, and all changes are also highlighted in the attached PDF “tracked version”. We hope you will find our revised manuscript acceptable for publication.

Reviewers’ comments:

Reviewer #1 (Comments for the Author):

The authors have satisfactorily responded to all my previous concerns. The conclusions raised are now strengthened by additional data and appropriate controls.

There are still a few typos that need to be corrected:

1. -line 173. I suppose the authors refer here to Figure S3 and not S2 (*in vitro* growth curve)

Authors’ response: First of all we are thankful to the reviewer for appreciation and acknowledging our revisions. We have changed the “Fig. S2” to “Fig S3D” in the revised manuscript (Line 174).

2. -line 301: "glutarnine". In addition, the Discussion is still very long (6 pages) and could certainly be shortened.

Authors' response: The 'glutarnine' has been modified to 'glutamine' in the revised manuscript (Line 219). Following your recommendation, we have shortened the 'discussion' section sharply.

Reviewer #2 (Comments for the Author):

1. The revision only very partially addresses my concerns and the writing of the manuscript still requires significant improvement. The LAM extraction/gel separation (now presented as Fig. 2A) has been improved, providing more convincing evidence for the LAM-deficient phenotype of the *embC* knock-outs and partial restoration in the complemented strain. Treatment of *M. abscessus* with ethambutol at high concentrations (32 mg/L) leads to inhibition of LAM synthesis supporting the inhibition of EmbC by EMB in this Mycobacterium species, albeit at concentrations higher than those used in the *in vitro* drug synergism study (Table 2). However, how this experiment was performed is neither described in the legend of figure 2 or in the Materials and Methods. When was EMB added? For how long? These are critical details that need to be added.

Authors' response: We appreciate your suggestion, so following your recommendation we have modified the 'Extraction and analysis of LAM' in the 'Materials and Methods' section to add the details regarding the extraction of LM and LAM in the revised manuscript (Lines 401-403). It has been described as, "The *M. abscessus* strains in logarithmic growth phase ($OD_{600} = 0.6$ to 1.0) were diluted 100 times to sub-culture for 24 h. EMB existed in broth for subculture at the final concentrations ($\mu\text{g/mL}$) of 0, 4, 8 and 32, respectively."

2. EMB shows no synergism with other antibiotics *in vivo* (Fig. 3C). The *embC* mutant is more susceptible to a number of drugs in 7H9 medium and *in vivo* (Table 1 and Fig. 3A) but to what extent the hypovirulence of the mutant contributes to this phenotype *in vivo* remains unclear. No clear pathogenicity data (including pathology and AFB staining) are shown. Again, the outcome of treating pharmacologically an established *M. abscessus* murine infection with an EmbC inhibitor may be very different from treating mice infected with the *embC* knock-out mutant which is totally

deficient in EmbC activity from the onset of infection and attenuated for virulence and pathogenicity. Therefore, the conclusions of the study remain rather speculative and do not significantly add to the field beyond what was already known of the hypersusceptibility of *embC* mutants to antibiotics in rapidly growing mycobacteria.

Authors' response: Thank you very much for your insightful comments. We recognize the limitations of our research. The contribution of virulence to drug sensitivity was not deeply studied. This is a very interesting scientific subject, which need to be studied further in the future. The main purpose of the current study was to explore the role of *embC* in intrinsic drug resistance of *M. abscessus* but not the virulence nor the pathogenicity. Our results (Fig. 3B) showed that bacterial burdens in lungs of the mice treated with CMC-Na (solvent control) in the knockout strain ($Mab^{\Delta embC}$) and the complementary strain ($Mab^{\Delta CMab}$) groups at the treatment completion were both significantly lower than that of at treatment initiation, which indicated that both $Mab^{\Delta embC}$ and $Mab^{\Delta CMab}$ were hypovirulent. In other words, the complementation did restore its virulence obviously. However, no drug killed $Mab^{\Delta CMab}$ in mice significantly, which provided evidence that linezolid (LIN), imipenem (IMP) and rifabutin (RFB) killed $Mab^{\Delta embC}$ in mice at least mainly not due to the virulence attenuation. Another evidence for the hypersusceptibility of $Mab^{\Delta embC}$ to LIN, IMP and RFB in mice mainly not due to the contribution of virulence attenuation is that vancomycin (VAN), clofazimine (CLF) and clarithromycin (CLA) as controls here did not lower the bacterial burden compared to the CMC-Na control, even though they became much more active *in vitro*. Therefore, we did not carry out study on pathogenicity including pathology and AFB staining in the current study. Even with such data, we could image that how we could explain the role of virulence attenuation. This need to be studied further by experts in this field. We totally agree with the reviewer that testing $Mab^{\Delta embC}$ vs testing $Mab^{wt} + EmbC$ inhibitor can be different. We have modified our conclusion to emphasize that this study just provided a clue to find potential EmbC inhibitor(s) which may enhance the activities of other drugs. Our study demonstrated for the first time that *M. abscessus* became hypersensitive to a variety of drugs *in vivo* after knockout of *embC* whereas, no similar *in vivo* studies have been published so far. We

also explained for the first time that EMB can enhance the sensitivities of multiple drugs by inhibiting LAM synthesis through EmbC in *M. abscessus in vitro*. In addition, there are many differences between *M. abscessus* and other fast-growing mycobacteria. For example when it comes to drug susceptibility, *M. abscessus* is intrinsically resistant to EMB, however, most fast-growing mycobacteria are sensitive.

3. The discussion was not shortened. It is actually longer than in the previous version of the manuscript. It is still unfocused and not clearly written with contradictory statements on the impact of pathogenicity on drug susceptibility (p. 17 , lines 329-336) and in other places as well (e.g., lines 267-269).

Authors' response: We acknowledge your concern therefore we have revised the 'Discussion' section (Lines 296-311) to make it concise and discussed more on the impact of attenuation on drug susceptibility, which may cause more attention of researchers in the related field. The sentence (Lines 267-269) has been removed in the revised manuscript.

4. The murine model used in the study remains poorly characterized. A reference is now cited to justify the lack of increase in bacterial burden between day 1 and day 10 but it's completely unrelated to the dexamethasone treatment model used here and, therefore, doesn't add anything to the discussion.

Authors' response: We have modified this part in the 'Discussion' section and to clarify the validity of our results we have cited another relevant study (Lines 284-288) in which they also used dexamethasone-induced immunosuppression mice. Now it is stated as, "The Mab^{Wt} load did not increase after 11 days of infection even in the dexamethasone (DEXA) -induced immunosuppression mice. A recent study showed that the *M. abscessus* burden in the lungs was unchanged after 1 week of infection using a similar DEXA-induced mouse model, but increased slightly after two weeks (27). This phenomenon indicates that it may need a longer period to observe a persistent uptrend in *M. abscessus* lung CFUs even in immunosuppressed mice."

Other concerns:

5. p. 10, line 172: Fig. S3 (not S2)

Authors' response: We have changed “Fig. S2” to “Fig. S3D” in the revised manuscript (Line 173).

6. p. 10, line 183: complementation (not "completion")

Authors' response: We have changed “completion” to “complementation” in the revised manuscript (Line 185).

7. p. 11, lines 209-210: What is a "nightmare" bacterium? Delete "nightmare"

Authors' response: We have deleted “nightmare” in the revised manuscript (Line 211).

8. p. 12: the reason for the lack of killing activity of EMB against *M. abscessus* should be explained in light of the sequencing of *embB* in the same isolate.

Authors' response: We have added the explanation of *M. abscessus* is resistant to EMB in the revised manuscript. It has been described as, “*M. abscessus* has been reported naturally resistant to EMB, which was supposed be due to amino acid difference in EmbB (another target of EMB) between *M. abscessus* and *M. tuberculosis* (25). The MICs of EMB against the selected *M. abscessus* strains exceeded to 64 µg/mL due to substitution of glutamine at 281 position instead of isoleucine (I281Q) in EmbB in all isolates in this study (Fig. S4), as has been reported previously (25).” (Lines 215-220).

9. p. 13, lines 251-253: EMB may not penetrate inside *M. abscessus* enough to reach its target.

Authors' response: Following your recommendation, we have revised this part in the revised manuscript (Lines 256-257).

10. p. 15, lines 278-279: "disruption" of what exactly?

Authors' response: The sentence has been removed in the revised manuscript.

11. Fig. 1: Typo (Tigecycline)

Authors' response: We have changed “Tegacyclin” to “Tigecycline” in the revised manuscript.

Thanks again to the both reviewers for critical review and constructive suggestions.

Yours sincerely,

Tianyu Zhang, Ph.D, Professor,

State Key Laboratory of Respiratory Disease,

Guangdong-Hong Kong-Macao Joint Laboratory of Respiratory Infectious Disease,

Guangzhou Institutes of Biomedicine and Health (GIBH),

Chinese Academy of Sciences (CAS).

July 18, 2022

Prof. Tianyu Zhang
Guangzhou Institutes of Biomedicine and Health, Chinese Academy of Sciences; University of Chinese Academy of Sciences
State Key Laboratory of Respiratory Disease
Room 207, 190 Kaiyuan Rd
Science Park
Guangzhou, Guangdong 510530
China

Re: Spectrum02763-21R2 (Arabinosyltransferase C Mediates Multiple Drugs Intrinsic Resistance by Altering Cell Envelope Permeability in *Mycobacterium abscessus*)

Dear Prof. Tianyu Zhang:

Your manuscript has been accepted, and I am forwarding it to the ASM Journals Department for publication. You will be notified when your proofs are ready to be viewed.

Sincerely,

Digby Warner
Editor, Microbiology Spectrum
